# Exploring the Link Between Vitamin K and Depression: A Systematic Review

**DOI:** 10.3390/medicina61050861

**Published:** 2025-05-07

**Authors:** Mohamad Hisham Hashim, Nik Nasihah Nik Ramli, Siti Nur Atiqah Zulaikah Nasarudin, Maisarah Abdul Mutalib, Muhammad Najib Mohamad Alwi, Aswir Abd Rashed, Rajesh Ramasamy

**Affiliations:** 1School of Graduate Studies, Management and Science University, Shah Alam 40100, Malaysia; 012025021319@sgs.msu.edu.my (S.N.A.Z.N.); maisarah_abdulmutalib@msu.edu.my (M.A.M.); 2International Medical School, Management and Science University, Shah Alam 40100, Malaysia; muhd_najib@msu.edu.my; 3Nutrition, Metabolism and Cardiovascular Research Centre, Institute for Medical Research, National Institutes of Health, Ministry of Health, Malaysia, No.1, Jalan Setia Murni U13/52, Seksyen U13 Setia Alam, Shah Alam 40170, Malaysia; aswir@moh.gov.my; 4Department of Pathology, Faculty of Medicine and Health Sciences, Universiti Putra Malaysia, Serdang 43400, Malaysia; rajesh@upm.edu.my

**Keywords:** vitamin K, depression, oxidative stress, osteocalcin, neuroinflammation

## Abstract

*Background and Objectives:* Depression is a multifactorial mental health disorder involving inflammation, oxidative stress, neuroplasticity deficits, and metabolic dysfunction. Emerging research suggests that vitamin K, beyond its classical roles in coagulation and bone metabolism, may influence neurobiological processes relevant to mood regulation. This systematic review evaluates the association between vitamin K and depressive symptoms and explores potential underlying mechanisms. *Materials and Methods:* A systematic search was conducted across PubMed, Scopus, Cochrane Library, ScienceDirect, and Google Scholar, following PRISMA 2020 guidelines. Eligible studies included human or animal research examining associations between vitamin K status (dietary intake or serum levels) and depression-related outcomes. Fourteen studies met the inclusion criteria: eleven observational studies, one randomized controlled trial (RCT), and two preclinical animal studies. *Results:* Most observational studies reported an inverse association between vitamin K intake or serum levels and depressive symptoms across diverse populations. One small RCT demonstrated modest improvements in depression scores following vitamin K2 (menaquinone-7) supplementation in women with polycystic ovary syndrome. Two preclinical studies using non-depression models reported behavioral improvements and reduced oxidative stress following vitamin K2 administration. *Conclusions:* While preliminary findings suggest a potential role for vitamin K in pathways relevant to depression, the current evidence is limited by cross-sectional designs, lack of isoform-specific analyses, and the absence of depression-focused preclinical models. Mechanisms including inflammation reduction, oxidative stress modulation, sphingolipid regulation, and vitamin K-dependent protein signaling (e.g., GAS6 and osteocalcin) were discussed based on indirect evidence and require further investigation in depression-specific contexts.

## 1. Introduction

Depression is one of the most prevalent and debilitating mental health disorders worldwide, affecting millions of individuals. According to recent estimates from the World Health Organization (WHO), approximately 280 million people suffer from depression. In 2021, an estimated 3.8% of the global population experienced depression, including 5% of adults—4% among men and 6% among women—and 5.7% of individuals over the age of 60 years [1]. Furthermore, depression is a significant concern among women during pregnancy and postpartum, with more than 10% experiencing depressive episodes [2]. Alarmingly, more than 700,000 people die by suicide annually, making it the fourth leading cause of death among individuals aged 15–29 years [1]. Despite the availability of effective treatments, over 75% of people in low- and middle-income countries receive no intervention, underscoring the urgent need for alternative therapeutic approaches [3].

Depression is characterized by persistent sadness, fatigue, cognitive impairments, and a loss of interest in daily activities [4]. It is a leading cause of disability and contributes significantly to the global burden of disease. Its etiology is complex and multifactorial, involving biological, genetic, psychological, and environmental factors. Conventional treatments, such as pharmacotherapy and psychotherapy, often have limited effectiveness, particularly in cases of treatment-resistant depression. As a result, there is growing interest in alternative and complementary interventions, including dietary modifications and micronutrient supplementation, natural product-derived compounds with potential health-promoting properties [5,6,7,8].

Among the micronutrients under investigation, vitamin K has emerged as a promising candidate in the context of mental health. Traditionally recognized for its role in blood clotting through the activation of clotting factors, vitamin K has also been implicated in a range of physiological processes beyond coagulation [9]. It plays a crucial role in bone health by activating osteocalcin and matrix Gla-protein, essential for bone mineralization. Additionally, vitamin K possesses significant anti-inflammatory and antioxidative properties, which are vital for cellular health and the regulation of oxidative stress. These functions are particularly relevant in depression, as chronic inflammation and oxidative imbalance are increasingly identified as key contributors to its pathophysiology [10,11].

Vitamin K exists in two main forms: vitamin K1 (phylloquinone) and vitamin K2 (menaquinones, MKs), which differ in their chemical structures, tissue distribution, and biological functions [12]. Vitamin K1, primarily found in green leafy vegetables, is mainly involved in the hepatic activation of clotting factors and has limited bioavailability in extrahepatic tissues. In contrast, vitamin K2, which is found in fermented foods and animal products, exhibits greater bioavailability outside the liver and exists in multiple subtypes (e.g., MK-4, MK-7) that vary in half-life and tissue distribution. Notably, MK-4, a subtype of vitamin K2, is present in high concentrations in the brain and has been implicated in neuronal health, sphingolipid metabolism, and neuroprotection. Vitamin K is also known to assist in the production of gamma-carboxyglutamic acid protein known as osteocalcin, which has been largely associated with bone health [13].

However, to date, no systematic review has comprehensively synthesized the available evidence on the relationship between vitamin K and depression. It remains unclear whether a consistent association exists, what types of populations have been studied, and which mechanisms may underlie this potential link. The aim of this review is therefore to identify, collate, and critically evaluate studies that explore this relationship. By incorporating findings from observational studies, clinical trials, and preclinical models, we seek to map the current state of research, highlight emerging mechanistic insights, and inform future directions in nutritional psychiatry. This exploratory synthesis is intended to establish whether a credible association exists and to describe the biological pathways that may explain the role of vitamin K in depression.

## 2. Materials and Methods

This systematic review was conducted in accordance with the PRISMA 2020 guidelines to ensure rigor and transparency throughout the process [14]. The review aimed to identify and synthesize high-quality evidence related to vitamin K and depression. The methodology comprised four key stages: identification, screening, eligibility assessment, and inclusion (Figure 1).

### 2.1. Search Strategy and Identification

A comprehensive literature search was performed across five electronic databases: PubMed, Scopus, Cochrane Library, ScienceDirect, and Google Scholar. The search strategy incorporated combinations of the following keywords: “Vitamin K”, “depression”, “major depressive disorder”, “menaquinone”, “phylloquinone”, “Vitamin K1”, “Vitamin K2”, “Vitamin K3”, “MK4”, and “MK7”. The initial search yielded 40 unique records, distributed as follows: Google Scholar (n = 8), PubMed (n = 13), ScienceDirect (n = 2), Cochrane Library (n = 0), and Scopus (n = 17).

### 2.2. Screening and Eligibility Criteria

Titles and abstracts of the 40 identified records were screened for relevance. During this process, 21 duplicate records were removed. Two additional records were excluded because they were non-original research articles, specifically letters to the editor. Full-text articles were then retrieved for detailed eligibility assessment.

#### Inclusion and Exclusion Criteria

Studies were included if they were original human observational studies, randomized controlled trials, or preclinical animal studies that individually assessed vitamin K intake or serum levels in relation to depression or depressive symptoms and employed validated depression assessment tools. Only peer-reviewed articles published in English were considered eligible. Studies were excluded if vitamin K was analyzed only as part of a broader nutrient pattern without specific isolation, if the study design was a machine learning exploratory model without direct hypothesis testing of vitamin K’s relationship to depression, or if the study explored other specific psychiatric diseases than depression.

Subsequently, 19 full-text articles were assessed for eligibility. During this assessment, three studies were excluded: two studies evaluated vitamin K only within broader nutrient patterns without individual vitamin K analysis, and one study used a machine learning exploratory model without directly testing the association between vitamin K and depression. No studies were excluded due to language barriers or inability to retrieve the full text.

### 2.3. Study Inclusion

Following full-text assessment, a total of 14 studies met the inclusion criteria. The eligible studies investigated the association between vitamin K, through either dietary intake or serum measurement, and depression or depressive symptoms, and used validated tools to assess depression outcomes. These 14 studies formed the basis of the final synthesis, comprising 11 observational studies, 1 RCT, and 2 preclinical animal studies.

### 2.4. Risk of Bias Assessment

The risk of bias for each included study was evaluated according to the study design. Randomized controlled trials were assessed using the Cochrane Risk of Bias tool 2.0 (ROB 2.0). Observational studies with control groups were evaluated using the Newcastle–Ottawa Scale (NOS), while preclinical animal studies and observational studies without control groups were assessed using the Agency for Healthcare Research and Quality (AHRQ) checklist. Most studies were rated as having a low risk of bias. However, five studies were judged to have a moderate risk of bias or some concerns, primarily due to limitations in comparability, methodological transparency, or incomplete blinding. A detailed summary of the risk of bias assessments is presented in Table 1. This review was not prospectively registered in a database such as PROSPERO, and no separate protocol was published prior to conducting the review. The identified risk of bias across studies was taken into consideration during the synthesis and interpretation of findings, with particular caution applied to studies assessed as having moderate risk.

## 3. Results

The initial database search identified 40 potentially relevant articles. After removing 21 duplicates and screening titles and abstracts, 19 full-text articles were assessed for eligibility. Of these, 14 studies met the predefined inclusion criteria and were included in the final synthesis. These comprised 11 observational studies, 1 RCT, and 2 preclinical animal studies. The studies selected are detailed in Table 2.

The included studies varied in design, population, and methodology. Observational studies were conducted across diverse demographic groups, including children, adults, older adults, pregnant women, and individuals with chronic health conditions, such as polycystic ovary syndrome (PCOS) and chronic kidney disease (CKD). Sample sizes ranged from 59 to over 30,000 participants. Vitamin K intake was assessed primarily through dietary recalls or food frequency questionnaires, while serum levels were measured using enzyme-linked immunosorbent assay (ELISA) or liquid chromatography–mass spectrometry (LC-MS/MS). Depression was evaluated using validated tools, including the Patient Health Questionnaire-9 (PHQ-9), the Center for Epidemiologic Studies Depression Scale (CES-D), the Geriatric Depression Scale (GDS), the Beck Depression Inventory-II (BDI-II), and clinical interviews based on the DSM-IV or DSM-IV-TR criteria.

Among the 11 observational studies, there was a consistent inverse association between vitamin K status and depressive symptoms. For instance, in a large cross-sectional study of U.S. adults using NHANES data, Zhang et al. [18] found that participants in the highest quartile of vitamin K intake had significantly lower odds of depression compared to those in the lowest quartile (odds ratio [OR] = 0.68; 95% confidence interval [CI]: 0.52–0.89; *p*-trend < 0.05). Similarly, Nguyen et al. [16] reported significant associations between lower dietary vitamin K intake and depressive symptoms among older Japanese adults, with ORs of 0.998 and 0.997 in men and women, respectively (*p* < 0.05). Hayashi et al. [21] observed a moderate inverse correlation between dietary vitamin K intake and CES-D scores in pregnant women (r = −0.496, *p* = 0.019). These findings remained statistically significant after adjusting for common confounders, including age, body mass index, comorbidities, and dietary quality. However, one study did not report on depression outcomes, instead noting an inverse relationship between vitamin K intake and anxiety levels in adults with CKD [25].

Experimental evidence from a randomized controlled trial by Tarkesh et al. [26] provided preliminary support for a causal relationship. In this study, women with PCOS who received 90 µg/day of vitamin K2 (MK-7) for eight weeks exhibited a significant reduction in depressive symptoms compared to those receiving placebo. BDI-II scores decreased from 16.9 to 15.0 in the intervention group, whereas scores slightly increased in the control group (from 13.8 to 14.0), with *p* < 0.05.

Two preclinical studies further supported the potential neurobiological role of vitamin K. Gancheva and Zhelyazkova-Savova [27] found that vitamin K2 supplementation significantly reduced depression- and anxiety-like behaviors in rats with diet-induced metabolic syndrome, as demonstrated by decreased immobility in the forced swim test and improved social interaction times (*p* < 0.05). Similarly, Mansour et al. [28] showed that vitamin K2 (MK-7) improved behavioral outcomes and reduced oxidative stress markers in ovariectomized female rats, a model of estrogen-deficiency-related mood disturbances.

Overall, the findings across human and animal studies indicate a consistent inverse relationship between vitamin K status and depressive symptoms. While the observational data suggest a robust association, causal inference is limited due to study design. Only one RCT was identified, and though supportive, further high-quality trials are needed to confirm efficacy and clarify the distinct roles of vitamin K1 versus K2. Additionally, considerable heterogeneity in population characteristics, outcome measures, and vitamin K assessment methods highlights the need for more standardized and targeted investigations.

## 4. Discussion

### 4.1. Insights from the Reviewed Articles

This review found a consistent inverse association between vitamin K status and depressive symptoms across diverse human populations. These associations, observed in large-scale observational cohorts and supported by preliminary causal signals from a small randomized controlled trial, suggest a biologically meaningful relationship rather than a chance finding. However, several critical gaps must be addressed before considering translational applications.

First, nearly all observational studies aggregated phylloquinone (vitamin K1) and menaquinones (vitamin K2) into a single exposure metric, obscuring key differences in isoform-specific bioavailability, tissue distribution, and the activation of vitamin K-dependent proteins such as Gas6 and osteocalcin. Given that MK-7, a form of vitamin K2, crosses the blood–brain barrier more efficiently and has a longer half-life than K1 [29], future studies must disaggregate isoforms to determine which molecular species are most relevant to neuroprotection and mood regulation. Second, the reliance on cross-sectional dietary recall data introduces recall bias and limits the ability to distinguish between acute and chronic vitamin K exposure. Integrating longitudinal biomarker panels—such as serum phylloquinone, MK-4/7 concentrations, and carboxylated osteocalcin—into cohort studies would help establish temporal relationships and clarify dose-response effects.

Although most studies reported statistically significant associations, the observed effect sizes were modest (ORs ~0.6–0.7), indicating that vitamin K is unlikely to function as a standalone antidepressant. Rather, it may act as a modulatory cofactor within a broader nutritional and metabolic framework. Interactions with other micronutrients (e.g., vitamins D and B), gut microbiome-derived menaquinone production, and individual genetic variations in vitamin K metabolism (e.g., VKORC1, GGCX polymorphisms) warrant systematic investigation. Future randomized trials should consider factorial designs that test vitamin K alongside established anti-inflammatory or neurotrophic agents. Concurrently, preclinical research should move beyond behavioral assessments to include molecular endpoints that are known to be implicated in depression pathophysiology.

In addressing the second aim of this review—exploring the underlying biological mechanisms linking vitamin K to depression—the available evidence remains limited. Only two preclinical studies met the inclusion criteria, both conducted in non-psychiatric disease models: one in rats with metabolic syndrome and the other in ovariectomized rats. Although both studies reported behavioral improvements with vitamin K2 supplementation, mechanistic insights were limited to general markers of oxidative stress. These included reductions in hippocampal malondialdehyde (MDA) and hydrogen peroxide levels, increased catalase (CAT) activity and total antioxidant capacity, and histological findings of preserved hippocampal pyramidal cell morphology, with minimal nuclear degeneration [28]. However, no studies directly examined central neurobiological mediators such as BDNF, TAM receptor activation, or neurotransmitter regulation. Thus, while these findings support the plausibility of a neuroprotective role for vitamin K, the current mechanistic evidence remains preliminary and indirect.

### 4.2. The Link: Potential Mechanism of Action (MOA) of Vitamin K on Depression

While the studies reviewed consistently suggest a link between vitamin K and depression, significant gaps remain in understanding the specific biological mechanisms underlying this association. To date, no preclinical studies have directly examined the effects of vitamin K in animal models of depression. Nevertheless, a broader neurobiological approach is warranted. Depression is increasingly recognized as a heterogeneous disorder involving multiple overlapping pathophysiological domains—including inflammation, oxidative stress, neurotrophic signaling, and myelin integrity—all of which are modulated by vitamin K in other disease contexts. The following mechanisms are therefore extrapolated from neurophysiological and disease models with shared features relevant to depression. These pathways should be interpreted as hypothesis-generating rather than definitive (Figure 2).

#### 4.2.1. Mechanism of Action 1: Antioxidant and Anti-Inflammatory Potential of Vitamin K

A growing body of evidence supports the antioxidant and anti-inflammatory roles of vitamin K in protecting the brain against oxidative stress and neuroinflammation. In vitro studies have demonstrated that vitamin K1 and MK-4 protect neural precursor cells from oxidative stress by reactive oxygen species (ROS) accumulation. This effect is partly mediated by the inhibition of 12-lipoxygenase (12-LOX), an enzyme involved in ROS production during lipid peroxidation [30,31]. Additionally, the antioxidant effect is enhanced through the activity of the vitamin K cycle, particularly the regeneration of vitamin K-hydroquinone, a potent radical-scavenging form of the vitamin that suppresses lipid peroxidation [32]. These findings highlight the role of vitamin K in modulating ROS-related damage through both enzymatic regulation and redox cycling.

In animal studies, Gancheva & Zhelyazkova-Savova [27] investigated vitamin K2 effects in rats with metabolic syndrome and found that vitamin K2 improved glucose metabolism and reduced anxiety and depression. Building on this, Chatterjee et al. have consistently explored its role across multiple rodent models of cognitive decline—induced by diabetes [33], D-galactose administration [34], aluminum chloride exposure [35], and gut dysbiosis [36]—concluding that vitamin K2 exerts neuroprotective effects. These studies collectively show that vitamin K2 enhances endogenous antioxidant systems (SOD, GSH, catalase, and NRF2) while mitigating neuroinflammatory responses through the downregulation of key proinflammatory markers such as TNF-α, IL-1β, MCP-1, NF-κB, and SIRT1.

Given that oxidative stress plays a significant role in the pathophysiology of both neurodegeneration and depression—largely through excessive ROS production and impaired antioxidant defense—vitamin K’s observed antioxidant and anti-inflammatory effects in preclinical models suggest potential therapeutic relevance. However, further research, particularly in clinical settings, is needed to confirm its efficacy and translational value in these conditions.

#### 4.2.2. Mechanism of Action 2: Vitamin K Regulation of Sphingolipid Biosynthesis

Vitamin K is essential for sphingolipid metabolism, supporting myelin integrity, neuronal function, and cognitive health. It regulates key sphingolipids like sulfatides and sphingomyelin, which are crucial for white matter stability, while also influencing gangliosides, which contribute to neuronal signaling. The enzymes serine palmitoyltransferase (SPT) and galactosylceramide sulfotransferase (GST) play central roles in this pathway and are influenced by vitamin K availability [37].

Studies show that higher MK-4 levels correlate with increased sulfatide concentrations in the hippocampus and cortex, regions vital for learning and memory, though this effect is region-specific [38]. In a demyelination animal model of multiple sclerosis, vitamin K supplementation enhances sulfatide production and supports myelin repair, highlighting its potential in remyelination therapies [39].

Beyond sulfatides, vitamin K—particularly its MK-4 form—has been associated with sphingomyelin biosynthesis, with deficiency leading to reduced levels of ceramides, sphingomyelin, and sulfatides, all critical for maintaining myelin integrity. An animal study showed that MK-4 concentrations are higher in myelinated brain regions and positively correlate with sulfatide and sphingomyelin levels, while showing a negative correlation with gangliosides [40]. Meanwhile, its deficiency has been linked to lower ganglioside levels, impaired cognition, and reduced exploratory behavior, reinforcing its role in neuronal communication [41].

Disruptions in sphingolipid metabolism have been implicated in various neuropsychiatric disorders, including major depression. Altered sphingolipid levels can impair synaptic transmission and neuronal plasticity—processes essential to mood regulation—while dysregulation may also compromise myelin integrity, contributing to the neurobiological changes observed in depressive disorders [42]. Given vitamin K’s role in sphingolipid metabolism, particularly in the biosynthesis of sulfatides and sphingomyelin, maintaining adequate vitamin K levels may support neuronal health and resilience against mood disorders. However, this mechanism has not yet been demonstrated in depression-specific models and requires further validation.

#### 4.2.3. Mechanism of Action 3: Vitamin K-Dependent Protein, GAS6

Vitamin K is essential for the γ-carboxylation of Growth Arrest-Specific 6 (Gas6), a vitamin K-dependent protein that interacts with the TAM family of receptor tyrosine kinases—Tyro3, Axl, and MerTK—to regulate various physiological processes, including those in the nervous system. While Gas6 can bind to all three receptors, it has the highest affinity for Axl, followed by Tyro3, and the lowest for MerTK [43].

In the brain, Tyro3 is predominantly expressed in neurons and oligodendrocytes, indicating a crucial role in neural function. Gas6-Tyro3 signaling has been shown to promote oligodendrogenesis and myelination in the adult central nervous system by enhancing oligodendrocyte generation and myelin production. These processes are vital for repairing demyelinating injuries, such as those seen in multiple sclerosis [44].

Although Axl and MerTK are part of the same receptor family, their functions in the brain differ from Tyro3. GAS6-Axl signaling plays a key role in neuronal survival and recovery after injury by regulating inflammatory responses and apoptosis. Additionally, this pathway is involved in microglial regulation, which is essential for maintaining central nervous system homeostasis [45].

GAS6-MerTK contributes to neuroprotection through distinct mechanisms, primarily by modulating neuroinflammation and supporting myelin homeostasis. Its activation suppresses the NLRP3 inflammasome via autophagy induction, leading to reduced neuroinflammation and improved neurological recovery following subarachnoid hemorrhage [46]. Additionally, GAS6-MerTK signaling inhibits Toll-like receptor-mediated inflammatory pathways in microglia, reducing pro-inflammatory cytokine production, while also playing a key role in myelin phagocytosis to support myelin maintenance and repair [47].

However, it is important to note that, while these mechanisms are well-supported in models of demyelination, injury, and neuroinflammation, they have not yet been demonstrated in depression-specific models. Future studies are needed to determine whether GAS6–TAM signaling plays a direct role in mood regulation or depressive pathophysiology.

#### 4.2.4. Mechanism of Action 4: Vitamin K-Dependent Protein, Osteocalcin

Vitamin K plays an essential role in activating osteocalcin, a hormone secreted by bone, by enabling its carboxylation through γ-glutamyl carboxylase—an enzymatic process necessary for osteocalcin’s calcium-binding function and biological activity. Emerging evidence from multiple lines of research points to a potential link between low osteocalcin levels and a wide array of psychiatric symptoms and disorders. Preclinical studies have shown that osteocalcin is capable of crossing the blood–brain barrier, where it contributes to several neurobiological processes critical for mental health, including enhanced learning and memory, increased neurogenesis, stimulation of monoamine neurotransmitter production, and balances GABA synthesis [48]. These processes are known to counteract mechanisms underlying various psychiatric conditions. Notably, osteocalcin-deficient mice exhibit pronounced anxiety- and depression-like behaviors, effects that are reversed upon the administration of osteocalcin [48]. Extending this work, Khrimian et al. [49] found that osteocalcin also promotes neuroplasticity by upregulating hippocampal BDNF expression and facilitating the intracellular transport of BDNF-containing vesicles, an effect mediated by the GPR158 receptor—further implicating osteocalcin in the modulation of mood and cognition.

Building on this, a recent study by Rnic et al. [50] provided human evidence by demonstrating both cross-sectional and longitudinal associations in youth: lower osteocalcin levels were linked to current psychiatric disorders and predicted greater increases in psychopathology symptoms over time. While these results point to a promising connection between osteocalcin and BDNF-mediated neuroplasticity, further research is necessary to confirm the role of vitamin K on these effects and assess their therapeutic potential for cognitive and emotional disorders.

## 5. Limitations

This systematic review has several limitations that should be acknowledged. First, the literature search was limited to studies published in English, which may have introduced language bias and restricted the inclusion of potentially relevant findings from non-English sources. This decision was made to ensure the accuracy of data interpretation, as automated or secondary translation tools may compromise the reliability of nuanced clinical data. Nonetheless, this may reduce the global generalizability of our findings, especially considering regional variations in dietary vitamin K intake and depression prevalence. In future updates, expanding the search to include non-English publications with appropriate translation support would help mitigate this limitation.

Second, many of the included studies did not differentiate between the two primary forms of vitamin K—phylloquinone (vitamin K1) and menaquinones (vitamin K2)—which may have distinct biological effects, particularly in the brain. While some experimental studies emphasized vitamin K2, most observational studies assessed total dietary intake or serum vitamin K without specifying the isoform. As a result, the relative contributions of K1 and K2 to mood regulation remain unclear. This represents an important gap in the literature that should be addressed through future studies focusing specifically on isoform-specific effects. Additionally, the majority of the included studies were cross-sectional in design, which limits causal inference and increases susceptibility to residual confounding. Although one randomized controlled trial and a few animal studies provided preliminary mechanistic support, these findings may not be fully generalizable to human populations. Sensitivity analyses were not conducted, which may limit the ability to assess the robustness of the findings across different subgroups or study designs. Certainty of evidence was not formally assessed using GRADE. Given the predominance of cross-sectional studies and variability in study designs, the overall certainty is likely low to moderate.

## 6. Conclusions

This systematic review synthesizes emerging evidence indicating a consistent inverse association between vitamin K status and depressive symptoms across diverse human populations. While most available data derive from observational studies, one randomized controlled trial and supporting preclinical research suggest that vitamin K—particularly menaquinones—may influence neurobiological pathways relevant to mood regulation. Proposed mechanisms include the modulation of oxidative stress, inflammatory signaling, sphingolipid metabolism, and neurotrophic support via vitamin K-dependent proteins such as GAS6 and osteocalcin.

However, key limitations constrain current interpretations. The majority of studies are cross-sectional in design, precluding causal inference, and often fail to distinguish between vitamin K isoforms. Mechanistic insights are derived primarily from non-depression models and remain hypothesis-generating rather than conclusive. As depression is increasingly recognized as a heterogeneous disorder with systemic underpinnings, vitamin K may represent a biologically plausible, upstream modulator within this broader pathophysiological framework.

To clarify its role in mental health, future research should prioritize longitudinal cohort studies with validated biomarkers, randomized trials stratified by vitamin K form, and targeted preclinical investigations using depression-relevant models. A more precise understanding of the neurobiological effects of vitamin K could ultimately support the development of novel nutritional strategies or adjunctive interventions in the prevention and treatment of depressive disorders.

## Figures and Tables

**Figure 1 medicina-61-00861-f001:**
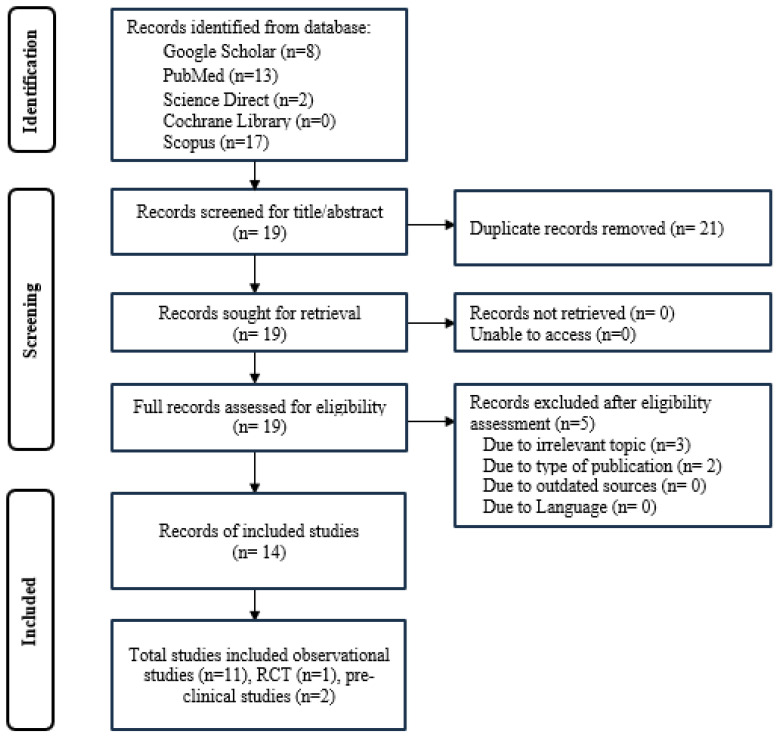
Prisma flow diagram.

**Figure 2 medicina-61-00861-f002:**
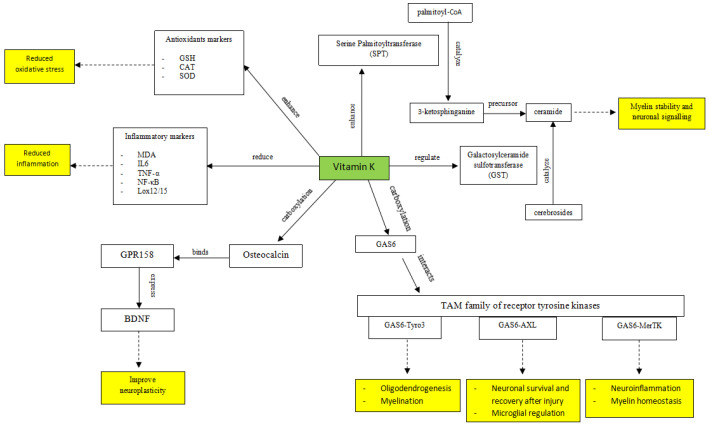
Potential mechanism of action (MOA) of vitamin K on depression.

**Table 1 medicina-61-00861-t001:** Risk of bias assessment table.

Study	First Author	Year	Study Type	ROB Tool Used	Overall Risk of Bias
1	Rubio-López et al. [15]	2016	Observational Study	Newcastle–Ottawa Scale	Low Risk
2	Nguyen et al. [16]	2017	Observational Study	Newcastle–Ottawa Scale	Moderate Risk
3	Bolzetta et al. [17]	2019	Observational Study	Newcastle–Ottawa Scale	Low Risk
4	Zhang et al. [18]	2023	Observational Study	Newcastle–Ottawa Scale	Low Risk
5	Wang et al. [19]	2024	Observational Study	Newcastle–Ottawa Scale	Moderate Risk
6	Wang et al. [20]	2024	Observational Study	Newcastle–Ottawa Scale	Low Risk
7	Hayashi et al. [21]	2020	Observational Study	AHRQ Checklist	Moderate Risk
8	Yu et al. [22]	2025	Observational Study	Newcastle–Ottawa Scale	Low Risk
9	Kaner et al. [23]	2015	Observational Study	Newcastle–Ottawa Scale	Low Risk
10	Khosravi et al. [24]	2019	Observational Study	Newcastle–Ottawa Scale	Low Risk
11	Mohtashamian et al. [25]	2024	Observational Study	AHRQ Checklist	Moderate Risk
12	Tarkesh et al. [26]	2022	Randomized Controlled Trial (RCT)	ROB 2.0	Moderate Risk
13	Gancheva & Zhelyazkova-Savova [27]	2016	Preclinical Animal Study	AHRQ Checklist	Low Risk
14	Mansour et al. [28]	2023	Preclinical Animal Study	AHRQ Checklist	Low Risk

**Table 2 medicina-61-00861-t002:** A systematic review analysis on vitamin K and depression studies.

Observational Studies						
No.	Study	Population	Depression Tool	Vitamin K Measure	Key Finding	Effect Size/Statistics
1	Rubio-López et al. [15]	Children (6–9 years), Spain, n = 710	CES-DC	3-day dietary recall	Lower vitamin K intake in children with depressive symptoms	*p* < 0.05 (between-group difference)
2	Nguyen et al. [16]	Adults ≥ 65 years, Japan, n = 1634	GDS	BDHQ (dietary intake)	Lower K intake associated with depressive symptoms	OR = 0.998 (m), 0.997 (f), *p* < 0.05
3	Bolzetta et al. [17]	Adults 45–79 years, USA, n = 4375	CES-DC	FFQ	Higher K intake linked to lower depression	β = −0.10, 95% CI: −0.17 to −0.03, *p* = 0.02
4	Zhang et al. [18]	Adults ≥ 18 years, USA, n = 11,687	PHQ-9	24-h dietary recall	Higher K intake inversely associated with depression	OR = 0.68, 95% CI: 0.52–0.89, *p*-trend < 0.05
5	Wang et al. [19]	Adults 18–80 years, USA, n = 30,408	PHQ-9	2 × 24-h dietary recall	Non-linear inverse association; mediated by oxidative balance	OR Q4 vs. Q1 = 0.64, 95% CI: 0.52–0.78, *p* < 0.001
6	Wang et al. [20]	Adults 18–65 years, China, n = 295	SCID	Serum vitamin K (LC-MS/MS)	Lower serum K in the suicide attempt group	OR = 0.61, 95% CI: 0.15–0.90, *p* = 0.004
7	Hayashi et al. [21]	Pregnant women, Japan, n = 776	CES-D	BDHQ	Higher K intake is associated with fewer depressive symptoms	r = −0.496, *p* = 0.019
8	Yu et al. [22]	Adults ≥ 18 years with CKD, USA, n = 5381	PHQ-9	24-h dietary recall + supplements	Higher K intake reduced depression risk	OR = 0.61, 95% CI: 0.44–0.85, *p* < 0.05
9	Kaner et al. [23]	Adults 18–60 years, Turkey, n = 59	DSM-IV-TR diagnosis	24-h dietary recall	Lower K intake in depressed patients	1353.9 mg vs. 2063.9 mg, *p* < 0.001
10	Khosravi et al. [24]	Adults 18–65 years, Iran, n = 330	DSM-IV/BDI	FFQ	Lower K intake associated with depression	OR = 0.99, 95% CI: 0.99–0.99, *p* < 0.05
11	Mohtashamian et al. [25]	Adults with CKD, Iran, n = 90	DASS-21	3 × 24-h dietary recall	Inverse correlation for anxiety; not reported for depression	β = −0.26, *p* = 0.015 (anxiety only)
**Clinical Trial**						
12	Tarkesh et al. [26]	Women with PCOS, Iran, n = 84 (RCT, 8 weeks)	BDI-II	Serum K (ELISA)	MK-7 improved depression scores vs. placebo	16.9 → 15.0 vs. 13.8 → 14.0, *p* < 0.05
**Pre-clinical Animal Studies**						
No.	Study	Model	Depression Measures	Key Finding	Effect Size/Statistics	
13	Gancheva & Zhelyazkova-Savova [27]	Male Wistar rats, metabolic syndrome model	FST, Social Interaction Test	K2 reduced depression-like and anxiety-like behavior	FST immobility reduced to 162.9 s, *p* < 0.05 vs. MS group	
14	Mansour et al. [28]	Female rats, OVX model	Modified FST, Open Field, Biochemical markers	K2 improved mood, behavior, and oxidative markers	All behavioral outcomes improved, *p* < 0.05 vs. OVX	

## Data Availability

This research does not contain new data.

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
