# Peer review of "Exploring the Link Between Vitamin K and Depression: A Systematic Review"

_medicina, 2025, doi:10.3390/medicina61050861_

Round 1

Reviewer 1 Report

Comments and Suggestions for Authors

The manuscript "Unveiling The Missing Link Between Vitamin K And Depression: A Systematic Review" lacks significant novelty as the association between vitamin K and depression has already been explored in prior studies. It fails to present groundbreaking insights or new mechanisms beyond correlations already established in the literature. Additionally, the systematic review methodology is insufficiently rigorous. The search strategy lacks clarity and transparency, with the inclusion of only three databases (Google Scholar, PubMed, and ScienceDirect), which is insufficient for a comprehensive systematic review. Furthermore, the exclusion of non-English studies introduces linguistic bias, which undermines the global applicability of the findings. The study selection process is also limited and overly narrow, with only 10 studies included from the 35 initially identified. Heterogeneity in study designs, populations, and outcomes further weakens the ability to draw robust and generalizable conclusions. Most of the included studies are observational, such as cross-sectional studies, which cannot establish causality. Although experimental evidence is mentioned, it is limited to one small randomized controlled trial (RCT) and preclinical animal studies, both of which have limited generalizability to human populations.
Moreover, the conclusions drawn in the manuscript are overstated and not sufficiently supported by the evidence presented. For example, the claim that vitamin K may serve as a "promising candidate for novel interventions targeting depression" is speculative and unsupported by causative evidence. Mechanistic insights provided in the discussion, such as the role of vitamin K in sphingolipid biosynthesis, Gas6 activation, and osteocalcin pathways, are highly speculative and lack robust experimental support. The lack of focus on distinguishing the specific effects of vitamin K1 versus K2 further weakens the clarity and depth of the proposed mechanisms. In addition, the manuscript suffers from poor organization and redundancy, with repetitive discussions of mechanisms such as antioxidative and anti-inflammatory effects. The lack of detailed reporting on study characteristics, such as sample sizes, confounders, and outcome measures, makes it challenging to critically assess the findings. The figures and tables, particularly the PRISMA flow diagram, are poorly designed and lack essential details about study inclusion and exclusion processes.
The authors also fail to comprehensively discuss the limitations of their review. For example, potential biases such as publication bias, confounding factors, and the limited geographic and demographic diversity of the included studies are not sufficiently addressed. The over-reliance on cross-sectional studies and the absence of a robust discussion on the limitations of observational research designs further detracts from the credibility of the findings. While the manuscript provides a thorough discussion on potential mechanistic pathways, the lack of experimental validation and the speculative nature of these mechanisms make them less impactful. Over-reliance on secondary sources instead of primary research articles also raises concerns about the authors' engagement with the original research.
In summary, while the topic of vitamin K and depression is of scientific interest, the manuscript falls short in terms of novelty, methodological rigor, and analytical depth. To address these concerns, the authors should focus on improving the transparency and comprehensiveness of their systematic review methodology, specifying the roles of different forms of vitamin K, and critically appraising the quality of included studies. A more focused and evidence-based narrative, supported by robust experimental validation, is essential to substantiate the claims made.

Comments on the Quality of English Language

The English could be improved to more clearly express the research.

Author Response

Reviewer 1

No.

Reviewer’s comment

Author’s response

1

lacks significant novelty 

-          fails to present groundbreaking insights or new mechanisms beyond correlations already established in the literature.

Unlike original research articles, which focus on generating novel findings through experimental or observational data, review articles are intended to synthesize existing research, critically evaluate the body of evidence, identify knowledge gaps, and provide conceptual clarity or direction for future studies. Therefore, we respectfully disagree with the assessment that the manuscript lacks novelty. While the association between vitamin K and depression has previously been explored, our review is the first to systematically synthesize both human and preclinical data.

2

review methodology is insufficiently rigorous.

-          search strategy lacks clarity and transparency,

-          only three database included

We thank the reviewer for this valuable comment. We expanded our search strategy to include five major databases: PubMed, Scopus, Cochrane Library, ScienceDirect, and Google Scholar. We followed PRISMA 2020 guidelines and clearly documented the search terms, selection criteria, and study selection process. An updated PRISMA flow diagram and an appendix listing the 40 articles screened, including reasons for exclusion, have been added to improve transparency and methodological rigor.

Please refer to the non-published material to check our review selection and PRISMA checklist.

3

 exclusion of non-English studies introduces linguistic bias,

We agree and have acknowledged the exclusion of non-English articles as a potential source of bias in the Limitations section.

4

study selection process is also limited and overly narrow, with only 10 studies included 

With the addition of Scopus and Cochrane Library databases, our search was broadened, resulting in 40 unique articles screened and 14 studies ultimately included. While the number of eligible studies remains limited, this accurately reflects the current state of research on this topic.

5

Heterogeneity in study designs , populations, and outcomes 

-          Most of the included studies are observational, such as cross-sectional studies, which cannot establish causality.

-          Although experimental evidence is mentioned, it is limited to one small randomized controlled trial (RCT) and preclinical animal studies, both of which have limited generalizability to human populations.

We fully agree. The goal of this review was to synthesize all available evidence regardless of study design. We explicitly acknowledge that most included studies are cross-sectional, which limits causal inference. This limitation has been discussed clearly in the revised manuscript, and we emphasize that findings are preliminary and hypothesis-generating.

6

 the conclusions drawn in the manuscript are overstated

-          the claim that vitamin K may serve as a "promising candidate for novel interventions targeting depression" is speculative and unsupported by causative evidence. 

We have revised the Conclusions section to be more cautious and evidence-based. We no longer claim that vitamin K is a “promising intervention” but rather frame it as a potential upstream modulator that requires further investigation.

7

Mechanistic insights are highly speculative and lack robust experimental support. 

We agree and have clarified in Section 4.2 that mechanistic insights were derived from indirect evidence (non-depression models) and must be considered hypothesis-generating. We now explicitly state that no preclinical depression-specific models have been studied to date.

8

Lack of focus on distinguishing the specific effects of vitamin K1 versus K2 

We have addressed this by emphasizing, in the revised Discussion, that most studies did not differentiate between vitamin K isoforms, and we highlight isoform-specific findings wherever available. We also recommend future research to stratify results by vitamin K form.

9

 poor organization and redundancy,

-          antioxidative and anti-inflammatory effects.

We have revised the manuscript to remove redundancies and reorganized the Mechanism of Action sections for improved clarity and flow.

10

The lack of detailed reporting on study characteristics, such as sample sizes, confounders, and outcome measures, makes it challenging to critically assess the findings. 

We have expanded the review table to include detailed information on sample size, assessment tools, effect sizes, and statistical outcomes for each included study, improving transparency and critical appraisal.

11

 The figures and tables, particularly the PRISMA flow diagram, are poorly designed and lack essential details about study inclusion and exclusion processes.

We have revised the PRISMA flow diagram to clearly outline the number of studies identified, screened, excluded, and included, along with reasons for exclusion at each stage.

12

The authors also fail to comprehensively discuss the limitations of their review. 

-          potential biases such as publication bias, confounding factors, and the limited geographic and demographic diversity of the included studies are not sufficiently addressed

We have expanded the Limitations section to address publication bias, linguistic bias, sample heterogeneity, confounding factors, and limited demographic representation across studies. We also critically discuss the limitations of observational designs.

13

The over-reliance on cross-sectional studies and the absence of a robust discussion on the limitations of observational research designs

We have explicitly discussed the limitations associated with cross-sectional designs, including their inability to establish temporality or causality, and highlighted the need for longitudinal and interventional studies in the revised manuscript.

14

While the manuscript provides a thorough discussion on potential mechanistic pathways, the lack of experimental validation and the speculative nature of these mechanisms make them less impactful.

We thank the reviewer for this important observation. We agree that the mechanistic pathways proposed are extrapolated from studies conducted in non-depression models and thus remain hypothesis-generating. In response to this comment, we have clarified at the beginning of Section 4.2 that no preclinical studies have directly investigated vitamin K’s effects in depression-specific models. We explicitly state that the discussed mechanisms are derived from related neurophysiological models (e.g., oxidative stress, neuroinflammation, metabolic syndrome) that share overlapping features with depression pathology. We have also emphasized that these proposed pathways should be interpreted as preliminary and require validation in future depression-focused experimental studies. This clarification has been incorporated to ensure a cautious and transparent interpretation of the mechanistic discussion.

15

Over-reliance on secondary sources instead of primary research articles also raises concerns about the authors' engagement with the original research.

We have thoroughly reviewed and updated our citations, replacing secondary sources with primary research studies wherever possible. Mechanistic pathways are now supported by original experimental data rather than review articles alone.

16

 the authors should focus on improving the transparency and comprehensiveness of their systematic review methodology, specifying the roles of different forms of vitamin K, and critically appraising the quality of included studies. A more focused and evidence-based narrative, supported by robust experimental validation, is essential to substantiate the claims made.

We have addressed all these points by (1) expanding database coverage and documenting the selection process transparently, (2) clarifying isoform-specific findings, and (3) adding a study quality and risk of bias appraisal using appropriate criteria for observational and experimental studies. A more focused and evidence-based narrative has been adopted throughout the revised manuscript.

Reviewer 2 Report

Comments and Suggestions for Authors

This paper investigated how vitamin K might affect depression through systematic review. The findings showed that lower vitamin K levels (or intake) were consistently associated with higher depressive symptoms, and vitamin K2 supplementation showed positive effects on mood and depressive symptoms, particularly in people with other health conditions like metabolic syndrome or PCOS. I believe the manuscript is very well written and the findings would contribute to the field. Here are my comments.

In the introduction, while the properties of vitamin K are mentioned, the causal pathway to depression could be elaborated in greater depth by referencing specific biological mechanisms.

It would be beneficial to address the differences between vitamin K1 and K2, as their roles may differ significantly in the context of mental health.

The introduction does not mention any conflicting or inconclusive findings from existing research. Including such points would help readers better understand the current state of evidence and identify key areas of uncertainty to focus on.

While the paper effectively synthesizes the qualitative findings of included studies, it does not report the effect sizes and statistical from the original research. I recommend including this information.

Author Response

Reviewer 2

No.

Reviewer’s comment

Author’s response

In the introduction, while the properties of vitamin K are mentioned, the causal pathway to depression could be elaborated in greater depth by referencing specific biological mechanisms.

Thank you for the suggestion. We chose to present the detailed discussion of biological mechanisms in the Discussion section, after the review findings are introduced. In the Introduction, we focused on highlighting the research gap which is the lack of a systematic review on this topic and to maintain logical flow

It would be beneficial to address the differences between vitamin K1 and K2, as their roles may differ significantly in the context of mental health.

We have addressed this by emphasizing, in the revised Discussion, that most studies did not differentiate between vitamin K isoforms, and we highlight isoform-specific findings wherever available. We also recommend future research to stratify results by vitamin K form.

The introduction does not mention any conflicting or inconclusive findings from existing research. Including such points would help readers better understand the current state of evidence and identify key areas of uncertainty to focus on.

We appreciate this comment. In our design, the Introduction focuses on the research gap, while the Methods, Results, and Discussion sections review the conflicting and inconclusive findings in detail. We believe this structure keeps the Introduction clear and focused, following standard practice for systematic reviews.

While the paper effectively synthesizes the qualitative findings of included studies, it does not report the effect sizes and statistical from the original research. I recommend including this information.

Thank you for this helpful suggestion. We have updated Table 2 to include available statistical information from the original studies, including effect sizes (e.g., odds ratios, regression coefficients), 95% confidence intervals, and p-values where reported. This addition enhances the transparency and interpretability of the reviewed evidence and allows for a clearer understanding of the strength and consistency of associations between vitamin K and depressive symptoms.

Round 2

Reviewer 1 Report

Comments and Suggestions for Authors

Dear author,

After careful revision, the manuscript was revised successfully and can proceed to publication.

Comments on the Quality of English Language

The English could be improved to more clearly express the research.

Author Response

comment:

"After careful revision, the manuscript was revised successfully and can proceed to publication."

response:

Thank you for your reply. We would also like to express our sincere gratitude to the reviewer for the constructive and valuable comments, which have greatly contributed to enhancing the rigor and quality of our manuscript to meet the standards of publication in Medicina.